# Genes Impacting Grain Weight and Number in Wheat (*Triticum aestivum* L. ssp. *aestivum*)

**DOI:** 10.3390/plants11131772

**Published:** 2022-07-04

**Authors:** Brandon J. Tillett, Caleb O. Hale, John M. Martin, Michael J. Giroux

**Affiliations:** Department of Plant Sciences and Plant Pathology, Montana State University, 119 Plant Biosciences Building, Bozeman, MT 59717-3150, USA; brandon.tillett@montana.edu (B.J.T.); caleb.hale@montana.edu (C.O.H.); jmmartin@montana.edu (J.M.M.)

**Keywords:** grain number, grain weight, wheat, source, sink, GNI, GW2, DA1, CKX6, GS5

## Abstract

The primary goal of common wheat (*T. aestivum*) breeding is increasing yield without negatively impacting the agronomic traits or product quality. Genetic approaches to improve the yield increasingly target genes that impact the grain weight and number. An energetic trade-off exists between the grain weight and grain number, the result of which is that most genes that increase the grain weight also decrease the grain number. QTL associated with grain weight and number have been identified throughout the hexaploid wheat genome, leading to the discovery of numerous genes that impact these traits. Genes that have been shown to impact these traits will be discussed in this review, including *TaGNI, TaGW2, TaCKX6, TaGS5, TaDA1, WAPO1,* and *TaRht1*. As more genes impacting the grain weight and number are characterized, the opportunity is increasingly available to improve common wheat agronomic yield by stacking the beneficial alleles. This review provides a synopsis of the genes that impact grain weight and number, and the most beneficial alleles of those genes with respect to increasing the yield in dryland and irrigated conditions. It also provides insight into some of the genetic mechanisms underpinning the trade-off between grain weight and number and their relationship to the source-to-sink pathway. These mechanisms include the plant size, the water soluble carbohydrate levels in plant tissue, the size and number of pericarp cells, the cytokinin and expansin levels in developing reproductive tissue, floral architecture and floral fertility.

## 1. Introduction

Since plant breeding’s inception, the primary goal has been to increase the agronomic yield of cereal crops utilized as major food sources. This is readily apparent in the history of common wheat (*Triticum aestivum* L.) improvement where the total production has increased globally by 0.9% from 1961 to 2008 [1]. These yield improvements primarily represent the genetic gains achieved via selection for adaptation to local environments but also include an increased production area. Continuing to increase the common wheat production area will become more difficult due to a scarcity of land suited to agriculture and changes in the rainfall patterns associated with climate change [2]. The percent of global agriculture acreage devoted to common wheat production has been as much as 21% [3]. In the 2020/2021 growing cycle, global wheat production totaled 776 million metric tons, produced across 223 million hectares (USDA 2022). The world’s population is projected to continue increasing, such that by 2050 it will be necessary to produce roughly twice as much food to sustain the global population as compared to 2010. This will require an improvement in yield averaging 2.4% per year [1], far above the current 0.9% increase per year for wheat. It is thus essential to focus on the identification of genetic variants that can be incorporated into new varieties that can increase wheat yield, since doubling the amount of arable acreage may not be possible.

When seeking to enhance the common wheat yield, plant breeders have been primarily focused on improvements to individual grain weight (GW) and grain number (GN). GW refers to the average weight of an individual grain and often appears in agronomic studies under the metric thousand kernel weight (TKW). GN represents the number of grains produced per wheat head. Agronomic yield is, therefore, a function of GW times GN times the number of productive heads over a specific area. There are numerous genes known to directly impact the number of productive heads within a specific unit area, such as *Teosinte branched 1* (*TB1*), which exhibits an effect on tillering, but they are not the focus of this review [4]. The focus here is on the genetic control of both the GW and GN. It is well documented that these two traits are negatively correlated, where increases in GN are associated with decreases in GW [5,6,7]; however, many studies have shown a transgressive segregation for yield when crossing high GN and high GW cultivars [8,9,10]. This trade-off is likely due to competition between grains as a sink, for a limited amount of source carbohydrates to “feed” the grain during grain fill [11]. The observation of transgressive segregation for yield when crossing high GN and high GW cultivars indicates there is room to improve the yield regardless of the negative relationship between these two traits. The trade-off between the GN and GW is an example of the dichotomy of the reproductive strategy and parental investment that exists in nature. Larger but fewer seeds represent more investment of resources for fewer individuals, and more numerous but smaller seeds represent less investment of resources into more individuals [5]. Studies focused on GN and GW across different cereals and environments have determined that GN is a highly plastic trait, while GW is significantly more heritable [5].

The following review will characterize some of the known genes that effect GW and GN, but it is necessary to point out that GW, GN, and yield are all polygenic traits. Many of the genes discussed in this review were shown to influence either the GN or GW through recombinant inbred line (RIL) experiments, only to find that they had no effect in subsequent experiments in different environments. Recent studies have demonstrated this genotype by environment interaction by repeating RIL experiments on a variety of known genes and quantitative trait loci (QTL) that effect the GN or GW, only to find significant results for less than half of the previously identified genes or QTL [12,13,14]. This does not indicate that the genes discussed are not useful to increase yield, but rather that the magnitude of the effect will be different under different environments and in different genetic backgrounds. A recent analysis of 390 diverse cultivars revealed that a significant majority of the yield increase in modern wheat is due to increasing both the GN and the number of wheat heads per area, with little gains in GW [15]. Given that most of the past improvement has been the result of increasing GNs, there is still a large amount of potential yield increase via increasing the GW. An analysis of 27 elite cultivars, grown in Valdivia, Chile, Ciudad Obregon, Mexico, and Colorado, USA showed a much smaller association between yield and GN in the USA as compared to Mexico and Chile, indicating that increasing the GW may be a more stable goal for yield improvement [16]. There are further implications here with respect to genotype by environment interactions, with environments that favor wheat (warmer, wetter, and longer growing seasons) garnering better yields through a focus on increasing the GN, while less favorable environments (cooler, drier, shorter growing seasons) will see more stable yield improvements through a focus on increasing GW.

Three major mechanisms of control over the GW and GN will be discussed in this review, the first of which will focus on how plant size and the available carbohydrates affect the traits, addressing the source end of the previously mentioned trade-off. This is followed by an exploration of the genes that influence the size and number of pericarp tissue cells at the time of anthesis and their relationship to GW. Finally, genes that influence the floral architecture, floret fertility and their relationship to GN will be discussed. Given that most past yield improvements were the result of increases in GN per unit area, the genes that primarily affect GW will be the focus. Genes impacting the GW include a serine carboxypeptidase, *TaGS5,* an E-3 ligase, *TaGW2*, a ubiquitin receptor, *TaDA1*, a cytokinin oxidase/dehydrogenase, *TaCKX6*, and an expansin, *TaExp6* (Table 1). An F-Box gene, *WAPO1*, and a transcription factor, *TaGNI*, that both affect the GN will be discussed as well (Table 1). Primers necessary to sequence the genes in this review are included in Appendix A. These genes were chosen because they each impact different plant development functions, alterations of which can be seen as different strategies for yield improvement. This review seeks to highlight how each of these strategies affects the source or sink tissue and makes recommendations for how agronomically preferable alleles of these genes might work best together to improve yield.

## 2. Plant Mass/Available Carbohydrates

The process of grain filling is informed by the well-established source-to-sink pathway that exists in plants. Plants utilize photosynthesis to build sugars which are in turn mobilized to the tissues in the plant for use in either new growth or are stored in seeds. The photosynthetic tissues represent the source while the grains represent the sink. In order to determine whether wheat GW was source or sink limited, researchers removed spikelets from different positions in wheat heads at anthesis and observed how the GW was impacted. No significant difference was observed in GW between the plants with the spikelets removed and the plants left fully intact [49]. The consistent GW measured between the plants with and without the spikelets removed suggests that there is not sufficient competition between grains for the available assimilates to limit their potential size. The implication is consistent with the fact that most yield enhancements over the past century have come by increasing the GN and reinforce the idea that GW is more sink than source limited. Subsequent repeat experiments across multiple environments showed the same results, indicating that the environment does not have a strong impact on wheat grain fill and that GW is more sink than source limited [50].

Regardless of the likelihood that grain fill is more limited by sink than source, there is evidence that increasing the source is associated with a higher GW. The source potential is largely made up of two different plant properties. First is the area of photosynthetically active tissue which directly impacts the amount of sugars the plant can produce. Second is the concentration of water soluble carbohydrates (WSC) that can be mobilized during grain fill as the photosynthetic tissues of the plant senesce [51]. While the environment can impact the WSC, an analysis of 116 diverse lines in four different environments revealed that differences in the stem WSC content were largely (90%) explained by genotype [52]. Researchers compared the GW of a RIL population, derived from the cross of a high and low WSC cultivar, finding a positive association between high WSC and GW [53]. The increased availability of WSC being associated with a higher GW contradicts the idea that source is not limiting for wheat during grain fill; however, the same study found that high WSC lines were associated with a smaller GN. There is a strong positive association between the photosynthetic material above the flag-leaf node and the GW in wheat [54], the implication being that enhanced source material leads to an enhanced GW. This suggests again that there may be some source limiting properties to grain fill. Early studies on the *Reduced Height 1* gene *Rht-1*, which is most well-known for controlling plant height, found a direct association between taller plants and a higher GW [55]. As the plants were reduced in height by mutations in different *Rht* homologues, so was their GW, indicating again that grain fill may be source limited. It is important to remember that while an increased plant size is associated with an increased GW, it is also associated with increased dry matter [55]. The increased dry matter, composed of longer stems, roots and more tillers and leaves, can be considered ancillary material that may be a competitive sink for grain development, especially under resource limited dryland conditions. Modern wheat breeding has frequently selected for mutant *Rht* genes to reduce plant height, which results in reduced individual GWs but enhanced yields [56]. This illustrates that while increased plant mass contributes to increased GW as well as ancillary tissues, the yields may also decrease.

Literature addressing the degree to which grain fill in wheat is source limited is contradictory at best. What can be definitively stated is that enhancing the source material, be it an increased photosynthetic flag-leaf area or higher WSC in the stem, is associated with a higher GW. Interestingly, there is an association between higher WSC lines and drought tolerance. It is theorized that under drought conditions grain fill is more dependent on the amount of WSC built up in the stem tissues made available during grain fill, as water limitations reduce the photosynthetic activity [57,58]. Conversely, high WSC are associated with an increased GW under irrigated conditions, likely due to the increased source for filling more numerous fertile florets tissues [53]. These findings have implications for breeding, indicating that enhancing the source materials used in grain fill may be a good strategy for seeking yield improvements in irrigated environments. Additionally, breeding for dryland environments will be favored by genes that increase WSC over time, supporting the grain fill under conditions of limited photosynthetic activity.

## 3. Controlling the Size of the Pericarp Tissue

Anthesis is the point at which pollen becomes viable, fertilization occurs, and anthers are extruded from the wheat head. This is an extremely important stage of development with respect to grain fill as there is a strong positive correlation between GW and the number and size of the pericarp cells at the time of anthesis [59,60]. Many genes have been demonstrated to impact the GW by controlling the size and number of pericarp cells at the time of anthesis through multiple biological processes. Some of the more thoroughly studied genes and the biological process they effect will be discussed in this section.

### 3.1. Serine Carboxypeptidase Gene Associated with GW–TaGS5

*OsGS5* is a gene that codes for a putative serine carboxypeptidase and is associated with changes in GW in rice (*Oryza sativa*) [61]. An overexpression of *OsGS5* in rice is associated with increased expression for five putative G1/S-phase genes involved in cellular mitotic division [61]. It was proposed that increased *OsGS5* expression promotes an increased mitotic division of cells through the positive regulation of important cell cycle genes, including the pericarp cells. *TaGS5* is orthologous to *OsGS5* and natural allelic variation for this gene has been associated with changes in wheat GW [17,18,19]. The *TaGS5* homeologues are located on chromosome group 3 [17]. For both the A and D homeologues, there are 10 exons and 9 introns over approximately 3700 bp, coding for a 482 amino acid serine carboxypeptidase. The B copy of *TaGS5* has significant structural differences, containing seven exons and six introns over approximately 15,000 bp and coding for an 875 bp amino acid putative serine carboxypeptidase [17,18,19]. An expression analysis showed similar patterns of expression for all the *TaGS5* homeologues (Figure 1). Each is abundant in younger developing seed tissue and drops off steadily during grain fill; however, differential expression levels were observed between the homeologues, with little expression from the B genome relative to more consistent expression levels from the A and D genomes [19].

Analysis of the coding regions of *TaGS5-3A* in 36 Chinese wheat cultivars revealed a polymorphism at the 2334 base pair position, where a T/G transition causes a missense mutation and an amino acid change of alanine to serine. The alleles are referred to as *TaGS5-3A-T* and *TaGS5-3A-G*, respectively (Table 2). Field trials of RILs that segregate for this polymorphism were performed in three environments over three years. The *TaGS5-3A-T* allele was associated with an 8% increase of GW over the *TaGS5-3A-G* allele. An enzyme activity assay determined that the total activity for the *TaGS5-3A-T* protein was approximately 50% greater than the activity of the *TaGS5-3A-G* protein, indicating that higher activity of this protein is associated with an increased GW [19]. A separate experiment found the same polymorphism when screening 41 land races as well as 322 modern Chinese cultivars. RIL experiments in two environments over three years observed a statistically significant 2% increase of GW for the *TaGS5-3A-T* allele as compared to the *TaGS5-3A-G* allele. Additionally, the *TaGS5-3A-T* allele was associated with higher expression levels in all tissues across six different time points in the wheat development as compared to the *TaGS5-3A-G* allele [17]. A polymorphism was also discovered and characterized in the promoter region through a screen of 40 modern Chinese cultivars. An insertion of a G at the −1925 position was associated with an increased expression of the *TaGS5-3A* gene and was labeled as *TaGS5-3A-b*, with respect to the wild type allele that lacked the insert and was labeled *TaGS5-3A-a* (Table 2). RIL experiments performed in one environment over three years demonstrated a 12.1% increase of GW for the *TaGS5-3A-b* allele as compared to the *TaGS5-3A-a* allele. Variety screening found this promoter mutation combined with the G/T SNP previously reported to form four different genotypes of *TaGS5-3A*. The allele with the promoter insertion and the T at position 2334 were associated with a 19.6% increase of grain weight in field trials over three years, making *TaGS5-3A-Tb* the most favorable allele for promoting a higher GW in any environment [18].

### 3.2. E-3 Ring Ligase Gene Grain Weight 2–TaGW2

The *OsGW2* gene influences rice GW and codes for a Ring-type E3 ligase which mediates ubiquitination by the proteasome system [20,21]. A loss of function of *OsGW2* leads to an increased rice grain cell number and higher GW, indicating that it is a negative regulator of GW [22]. *TaGW2* is an orthologue of *OsGW2* in wheat that shares 87% nucleotide homology [23]. An in vitro ubiquitination assay in which *TaGW2-A* was fused to a 6Xhis tag and mixed with E1 and E2, resulted in self-ubiquitination determining that *TaGW2* is an E3 ligase as observed in rice [24]. *TaGW2* has a coding region of approximately 1275 bp that spans eight exons with 98% homology in cDNA between the A, B, and D genomes and maps to the short arm of chromosome 6 [20,22]. The resulting protein from any of the three homologues is 424 amino acids in length [20]. The expression levels of *TaGW2* in seeds are relatively even across all three homologues [25] (Figure 2); however, the timing of the expression through development was differential with the A and D genomes contributing the most expression leading up to anthesis and the B genome contributing the most to expression during late grain fill [24]. When *TaGW2* is silenced with RNAi, the transcript levels of the cytokinin synthesis genes such as *TaIPT2* are increased along with decreased levels in the cytokinin degradation genes such as *TaCKX1* [26]. Cytokinin plays a role in the process of cell division and cell elongation in all plants and variable cytokinin levels have been shown to influence GW in rice and wheat [26,62,63]. It is proposed that the association between *TaGW2* and changes in GW are due to changes in cytokinin regulatory gene expression allowing for more cell division and pericarp expansion when the *TaGW2* expression is reduced.

Reducing the *TaGW2* by 50% using RNAi increased the GW by 4% [25]. More recent *TaGW2* RNAi knockdowns reduced *TaGW2* expression to approximately 15% of the wild type and increased the GW by 18% [27]. Thus, *TaGW2* acts as a negative regulator of GW, and alleles that are less functional or non-functional will increase the GW. There is also evidence that the effects of *TaGW2* mutations are additive. Field experiments utilizing TILLING or CRISPR-mediated mutants have shown that a single *TaGW2* knockout increases the GW by approximately 5%. Double knockouts increased the GW by approximately 11%, and triple knockouts increased the GW by approximately 18% [28]. In a separate TILLING experiment only on *TaGW2-6A*, mutants exhibited a 6.6% GW increase across multiple environments and years [29].

Natural allelic variation for *TaGW2* has been explored and characterized. An A/G transition in the *TaGW2-6A* promoter region at −494 bp was associated with GW *variation* (Table 3). The *TaGW2-6A-A* allele was associated with an increased GW and lower *TaGW2-6A* expression [30]. Two other SNPs in the promoter were associated with GW. Both were A/G transitions at positions −593 and −739 (Table 3). In RIL field experiments, the haplotype with a G at position −593 and an A at position −739 were associated with a 9.2% increase of GW as compared to the haplotype with opposite bases at those two positions, as well as a decrease in the transcript expression levels [64]. An insertion of a T at position 977 in the eighth exon creates a frameshift mutation and truncation of the last quarter of the *TaGW2-6A* protein (Table 3). This allele, *TaGW2-6A-T,* is considered a natural loss of function and is associated with a 9.0% increase of GW as well as decreased *TaGW2-6A* transcript levels [23,26]. Hundreds of cultivars from around the globe were sequenced to discover the alleles mentioned above and in no case do any of the alleles that decrease the TaGW2-6A expression and increase GW appear in more than 25% of the populations. This implies that there was little to no selection pressure on the *TaGW2-6A* alleles that increase GW during domestication (or that those alleles arose post-domestication), and that increasing the GW without reducing the GN may be possible by selecting for reduced TaGW2 function.

A possible explanation as to why the *TaGW2* reduced-expression alleles are not found in higher frequencies in modern cultivars may be due to this gene’s association with increased starch in grain filling. A decrease in *TaGW2* expression through natural allelic variation or induced knockouts were associated with increases in the expression of *TaAGPL* and *TaAGPS*, with both genes coding for a subunit of the AGPase heterotetrameric enzyme [26,27]. AGPase is involved in starch-biosynthesis and overexpression of the AGPase genes in *Zea mays* L. leads to an increase in the accumulation of starch in the kernel [65]; however, experiments in rice have yielded differential results in which transformations to overexpress AGPase in leaf and grains did not increase the GW or alter the starch percentage, but instead increased the GN [66,67]. Protein content is an important agronomic aspect of the wheat grain, and it is possible that increases in GW through the *TaGW2* locus could be predominantly recognized as increases to the starch reserves in the grain. A decreased expression of *TaGW2* leads to increases in GW, but those increases may be primarily from increased starch in the grain fill, perturbing the potential for exploitation of the *TaGW2* locus for stably increasing the GW in an agronomically useful manner. There has not been much analysis into the protein content of grain after manipulation of the *TaGW2* locus, but more explanation is warranted as the magnitude of the effect of *TaGW2* on GW is quite significant. Potentially, an increased AGPase expression in wheat associated with changes to TaGW2 may promote a higher grain number along with the increased GW effect of the gene. This could explain why some TaGW2 experiments observed the GN holding stable while the GW increased, rather than the expected decrease in GN often associated with an increased GW.

### 3.3. Ubiquitin Receptor–TaDA1

*TaDA1* was recently discovered to effect GW and interacts with *TaGW2* in yeast two-hybrid and firefly luciferase complementation assays [31]. *TaDA1* overexpression is associated with a decreased GW, while RNAi-mediated reductions in *TaDA1* expression increases the GW [31]. Interestingly, there were no changes in other important agronomic traits between wild type and transgenic lines, specifically in grains and spikelets per spike, suggesting that *TaDA1* downregulation would increase the GW without decreasing the GN. *TaDA1* TILLING-derived mutations were identified for each homologue, combined by crossing, and used to create RILs. Field experiments of this RIL population found an 8% increase of GW associated with the knockout of *TaDA1*; however, there was a reduction in GN in this experiment, such that the overall yield was not different between the allele classes [32]. This suggests that changes in *TaDA1* may not be a suitable tool in breeding for yield increases, or at least the genotype by environment interaction effects for this gene are not yet well understood.

*TaDA1* is located on chromosome set two, contains 11 exons spanning approximately 6400 bp and each of the homeologues are predicted to be translated into 504 amino acid proteins [31,32]. The expression of *TaDA1* is relatively consistent across all three genomes and across different plant tissues (Figure 3). In *Arabidopsis thaliana* (L.) Henyh., *DA1* has been shown to be a ubiquitin receptor that interacts directly with E3 ubiquitin ligases [33]. There is a large amount of homology between the *DA1* gene in wheat and Arabidopsis (approximately 75% at the protein level) and given the established interaction between *TaDA1* and *TaGW2* in wheat, it is assumed that the function is the same. Transformation experiments have revealed an increase in the pericarp tissue size and number when *TaDA1* is downregulated [31]. This suggests that *TaDA1* effects the GW by negatively regulating the proliferation of maternal cells, similar to *TaGW2*. RILs that segregate for allelic variation of *TaDA1* and *TaGW2* have demonstrated an additive effect on the GW [31]. An exploration of the proteome in independent lines with downregulated *TaDA1* and *TaGW2* revealed a large amount of overlap for the associated gene expression changes. Changes in the expression of *TaDA1* were associated with different levels of 95 separate proteins in vivo and changes in the expression of *TaGW2* were associated with different levels of 249 separate proteins, in which 46 proteins were common to both cases [31]. Both *TaDA1* and *TaGW2* act within their own regulatory pathways, but share some overlap in their function. It is suggested that the magnitude of the effect of *TaGW2 on GW* would be greater than that of *TaDA1* given the larger change to the proteome, which was demonstrated in a single experiment that looked at both genes in the same population [31].

Allelic variation at the coding level has been discovered in *TaDA1* in wheat; however, none of these changes were associated with significant changes in GW [31] (Table 4). Multiple SNP mutations in the promoter region have been characterized for both their effect on the transcript levels and on the GW. Consistent with what was observed in RNAi experimentation, the haplotype with the lowest levels of expression was associated with the highest GW [31]. Changes in the GW due to manipulation of the *TaDA1* locus may affect other important agronomic properties of the grain. Lines with knockout mutations aggregated in all *TaDA1* homeologues had an overall increased starch content of 2% as compared to the wild type [32]. Protein content is an important wheat quality metric and increasing GW through the downregulation of *TaDA1* may result in a lower than desirable protein percentage in the final grain.

### 3.4. Cytokinin Oxidase/Dehydrogenase Genes–TaCKX6

Changes to the cytokinin levels by *TaGW2* were proposed as a mechanism that effects GW; therefore, genes involved more directly in the cytokinin pathway may also affect the GW. In addition to playing a role in cell division, cytokinin can increase the content and stability of chloroplasts [26,34,62] and the result of an increase in chloroplasts represents an improvement to the source side of the source/sink relationship. Changes to the plant cytokinin levels may alter the GW by both altering the sink potential in the number of pericarp cells at the time of anthesis, as well as improving the source potential of the plant by increasing the photosynthetic production of sugars. *CKX* is a family of genes found in wheat known as cytokinin oxidase/dehydrogenases and their function is to permanently deactivate cytokinin [35]. Changes to the *CKX* genes impact many agronomically important wheat traits by mediating the amount of active cytokinin [36]. Different *CKX* genes are associated with different stages of development impacting both the source and sink potential of the wheat plant. Altering the source potential, *TaCKX10* is associated with root growth, *TaCKX3* is associated with tillering and *TaCKX5* is associated with the timing of the flag-leaf senescence [37]. *TaCKX1*, *TaCKX4* and *TaCKX6* have all been associated with GW and floret fertility, thus, altering the sink potential [37,38,39,63]. A negative correlation has been established between the expression levels of *CKX* genes and GW. A higher expression of *CKX* genes lead to a reduction in active cytokinin in the cell, which lead to a reduction in either the source potential (tissues for photosynthesis) or sink potential (fewer pericarp cells at the time of anthesis). Additionally, genotypes with multiple copies of *TaCKX4* are associated with increased expression levels and a decreased GW [63].

Nulli-tetrasomic lines have been used to reveal the location of *TaCKX6* on the chromosome group 3, and natural allelic variation has been discovered on the D genome copy [38,39]. *TaCKX6-D* has an open reading frame of 1638 bases spanning three exons and is predicted to translate to a 545 amino acid protein [39]. The natural expression of *TaCKX6* is found predominantly in reproductive spike tissue and grain with relatively double the expression from the A genome than the B and D (Figure 4). An 18-base pair indel was discovered in the second intron of a modern Chinese cultivar and labeled *TaCKX6-D1a.* Two years of RIL field experiments found and associated a 7% increase of GW for the line with the indel over the wild type [39] (Table 5). The expression of the wild type allele was between 1.5- and 5-fold higher than the *TaCKX6-D1a* allele containing the indel, supporting the assertion that *TaCKX6-D* is a negative regulator of GW [39]. A 29 base indel in the 3′ UTR, directly following the TGA stop codon, is also located on the D genome copy and labeled *TaCKX6-D1c* [38] (Table 5). *TaCKX6-D1c* was associated with a 43% increase of GW as compared to the allele lacking the wild type in one year of RIL field experiments [38]. The expression levels were not analyzed in the experiment on the *TaCKX6-D1c* allele, but the magnitude of the effect suggests that *TaCKX6-D* has a major influence on GW.

### 3.5. Expansin—TaExpA6

The plant cell wall consists of a network of polysaccharide chains that form a particularly strong and flexible container for the cell, which ultimately limits its size. Expansins play a role in manipulating the structure of the cell wall such that turgor pressure allows the cell to expand [68,69]. Given the established relationship between the number and volume of pericarp cells at the time of anthesis and GW, it makes sense that changes to the expansin levels could enhance the GW by promoting larger pericarp cells. *TaExpA6* transcribes an expansin that is found concentrated in roots as well as the pericarp cells of wheat during early development through anthesis [40]. Using the promoter for puroindoline-b, which drives expression in the developing endosperm and pericarp tissues, transgenic lines that overexpressed *TaExpA6* in those tissues were constructed. A 3- to 12-fold increase of *TaExpA6* expression in the pericarp cells is associated with increases of GW from 10.9 to 26.6% [41]. No changes to the GN were associated with changes to the *TaExpA6* expression in the pericarp cells, indicating that the manipulation of expansin-related genes may be a pathway to increasing the yield outside of the trade-off between the GN and GW. Studies have shown this association between an increased expression of *TaExpA6* and higher GW in haplotype analysis [42]. There are a predicted 128 different expansin genes located in the wheat chromosome, active in different tissues at different times, with a 75% or greater sequence homology between all of them [70]. A majority of the expansin genes contain six exons and code for a protein of approximately 250 amino acids with three conserved domains; a signal peptide, a *DPBB_1* domain and a *Pollen_allerg_1* domain. *TaExpA6* is located on chromosome set four. Allelic variation for *TaExpA6* is currently undiscovered; however, natural expression differences for *TaExpA6* would be a potential source for a stable GW improvement.

## 4. Floral Architecture and Floret Fertility

Much of the breeding efforts of the past towards increasing the yield have resulted in increasing wheat GNs per head, which is a function of both the floral architecture and floret fertility. The floral structure of wheat is frequently referred to as inflorescence architecture, where an inflorescence can be thought of as the meristem for floral tissue. After termination of the inflorescence growth occurs in the wheat plant, the expression of various floral development genes determines the final development of the inflorescence tissue into a specific type of flowering structure [71]. A delayed inflorescence termination can result in increased branching of the inflorescence tissues and has been associated with increased spikelet development in rice [43,71,72]. Directly following anthesis is a period where many flowering primordia in the wheat head undergo floret abortion, resulting in a reduction in the number of florets that will ultimately set grain by one half to one third of the number developing primordia during the green anther stage [73,74]. A study of 30 European winter wheat cultivars revealed an association between the genotype and the proportion of florets that undergo abortion, indicating genetic control over the trait [73]. Recently, genes have been discovered which directly impact the floral architecture and floret fertility and, in turn, the number of grains produced in the average spikelet. Some of these genes will be discussed in the following section.

### 4.1. Wheat Aberrant Panicle Organization–WAPO1

The Wheat Aberrant Panicle Organization 1 (*WAPO1*) gene codes for an F-Box protein [44]. In wheat, the F-box proteins function as a component of the Skp1–Cullin1–F-box (SCF) complex that recognizes substrates to be ubiquitin tagged for degradation [75]. The SCF complex interacts with E2 ligases which can ubiquitinate substrates for targeted degradation by the proteosome system. This is a separate pathway from *GW2*, an E3 Ring-type ligase discussed above; however, both pathways result in the ubiquitination of specific substrates for degradation by the proteosome. The specific substrates that the *WAPO1* F-box interacts with in wheat are unknown; however, the orthologous gene in rice, (*APO1*), has been shown to interact with C-class MADS box genes [43]. C-class MADS box genes are transcription factors known to be involved in floral tissue determination [76]. Experiments in hexaploid wheat, where *WAPO1* from the A genome was overexpressed, found an increase in the spikelets per spike [45]. It is theorized that *WAPO1* positively controls the spikelet number by delaying the termination of the inflorescence growth through the inhibition of floral determination genes. This delay of the termination of inflorescence leads to more branching before spikelet development, resulting in more spikelets. At higher levels of *WAPO1* expression, more spikelets were observed; however, the percent of infertile spikelets increased with the expression, such that the plants with the highest expression of *WAPO1* were associated with a decrease in yield. *WAPO1* contains two exons and one small intron, spanning approximately 1460 base pairs, and is predicted to translate into a 440 amino acid protein [44]. A single copy of *WAPO1* exists on each of the three chromosomes from set seven with a relatively even, but low expression across all three genomes (Figure 5). The expression of *WAPO1* is found primarily in the inflorescence meristem tissue of the developing wheat head.

Two major alleles have been characterized for *WAPO1* on the A genome. The *WAPO-A1a* allele differs from the *WAPO-A1b* allele by a 115 bp deletion located −599 to −485 upstream of the start codon as well as a SNP in the F-Box binding the motif region of the gene that changes the 47th amino acid from a cystine to a phenylalanine [46] (Table 6). A BLOSUM score of −2 is predicted for the amino acid change in the F-Box binding motif region indicating a likely decrease in protein functionality. In addition to these differences, there are three SNPs in the promoter region, one in the intron and a final SNP that changes the 384th amino acid from an aspartic acid to an asparagine [44]. None of these additional SNPs had a BLOSUM score that would suggest a decrease in function. HIF pairs that are homozygous for *WAPO-A1a* were found to have a three-fold lower expression than those that were homozygous for *WAPO-A1b* in qRT-PCR experiments [46]. Growth experiments utilizing hexaploid wheat RILs that varied for *WAPO-A1a* and *WAPO-A1b* found a 4.7% increase of GN per spike coupled with a 2.3% decrease of GW associated with *WAPO-A1b*, the more highly expressed allele. The yield was found to be 2.1% higher for the *WAPO-A1b* allele indicating that the benefit to an increased GN was not completely countered by the decrease in GW. The changes to GN per spike were significantly greater in CRISPR knockout experiments targeting *WAPO1*, indicating that the *WAPO-A1a* allele is a reduced-function allele and not a complete loss of function allele [44]. Separate experiments on NIL pairs that varied for the *WAPO-A1a* and *WAPO-A1b* alleles found an average 5.6% increase of GN per spike and a decrease of 2.8% of GW [77]. The yield was held stable between the genotypes for this set of experiments.

### 4.2. Grain Number Increase—GNI-A1

The Grain Number Increase (*GNI*) gene codes for an HD-Zip transcription factor in wheat [47]. The HD-Zip transcription factor class genes are unique to plants and contain a homeodomain for specific DNA binding and a leucine zipper which acts as a dimerization motif, allowing the protein to bind to specific sites in DNA and assist in transcription activation [78]. RNAi experiments have demonstrated a negative relationship between expression of *GNI-A* and floret fertility such that it is theorized that *GNI* promotes floret abortion [47]. *GNI* was identified from a QTL associated with GN located on the long arm of Chromosome set two. The gene contains three exons spanning approximately 1700 base pairs and is predicted to translate to a 218 amino acid protein. Three homeologues exist in most modern hexaploid wheat with one copy on each of the three genomes. Four wild emmer accessions from Israel possess a second copy of *GNI* on the A genome [48]. *GNI* expression is highest in the distal ends of spikelets during development leading up to anthesis. The expression level of *GNI* from the A and D genomes is confounded, with a recent study reporting the expression from the A genome being two- to five-folds higher than the D genome [47]. This is contradicted by information from the Wheat Expression Browser database, reporting a two- to three-fold higher expression from the D genome relative to the A genome (Figure 6). This could be due to the high sequence similarity (96%) between the A and D genomes in the cDNA, which can perturb the sensitivity of a transcript abundance analysis. Published literature and the data on Wheat Expression Browser agree that there is relatively no expression of *GNI* from the B genome.

Three separate alleles of *GNI-A* have been characterized, and a haplotype analysis of 111 diverse lines, ranging from wild emmer to cultivated durum revealed 11 different haplotypes based on 11 SNPs; however, only two of these haplotypes were identified in the cultivated durum [48]. These two modern haplotype groups consist of three alleles, all resulting from a SNP in the homeodomain coding region of the second exon. The wild type of this allele, *GNI-A105N*, codes for an asparagine at the 105th amino acid position in the protein, while the *GNI-A105K* allele codes for a lysine and the *GNI-A105Y* codes for a tyrosine [47] (Table 7). The transcript levels of the modern alleles of *GNI-A* were found to be lower than the ancestral alleles present in wild emmer [48]. This indicates that domestication has created selection pressure for reduced expression alleles of *GNI*. Field experiments utilizing RILs revealed a significant increase in GN for lines with the *GNI-105Y* mutant allele as compared to the *GNI-105N* wild type allele [47]. Interestingly, in this experiment there was no observed change in the GW, indicating that the manipulation of *GNI-A* could increase yield through stably increasing the GW without reducing the GN; however, RIL experiments that compared *GNI-105Y* to an ancestral allele of GNI, retrieved from wild emmer, found an increase in GN associated with *GNI-105Y* and a proportional decrease in GW, such that the yield was unchanged. Further experimentation on the RIL population involved the removal of distal spikelets, which resulted in an increased GW for the central spikelet-derived grains, indicating a potential sink limitation in the form of competition between grains for resources. The magnitude of this effect was greater in the lines containing the modern *GNI-105Y* allele relative to the higher expressed ancestral allele [48]. Breeding for an increased GN may make a plant that is otherwise sink limited become more source limited. This reveals that breeding for increased GN may be more effective in environments that have fewer source limitations, such as irrigated environments, while focusing on breeding for GW may be more appropriate for source limiting environments such as drylands.

### 4.3. Awnletted Wheat

A major varietal phenotype of the wheat head is the presence or absence of awns as well as an intermediate state called awnletted. Experiments spread out over 25 different environments in Australia and Mexico utilized near isogenic line (NIL) pairs that varied for the presence of awns. Awnless lines produced 5% more grains per spike as compared to awned lines; however, the awnless lines had a reduction of GW by approximately 5%, allowing for yields to remain constant across genotypes. Given that awns can make up as much as 40% of the biomass in the spikelet, it is theorized that awned lines provide a competitive sink with other tissues in the spikelet, resulting in the production of fewer fertile florets [79]. It has been shown experimentally that there is a positive relationship between larger awns and an increased GW [80]. Awns provide photosynthetic assimilates directly to developing grains during the filling stage as well as add additional thermoregulation to the tissues in the wheat head through increased evapotranspiration through stomata on the awn [81,82]. Stable yield increases may not be achievable by exploiting the trade off in GN and GW associated with awns, but the effect of awns on these traits are worth considering when breeding for individual environments. It may just be that a well irrigated and fertilized environment can produce plants with a large enough source potential that the benefit in GN for an awnless line may outweigh the detriment to the GW. Conversely, an awned line may be more appropriate for a dryland environment, where source limitations favor lines with fewer but larger grains.

## 5. Discussion

The first and foremost goal of breeding for globally utilized cereal grains has been and remains increasing the yield. Historically, efforts to improve wheat have focused on improving the yield, which has been predominantly accomplished by increasing the GN while keeping the GW constant, or vice versa [15]. Frequently, breeding efforts to increase one trait result in a decrease of the other, while the yield itself is held constant [5,6,7]. This is largely due to the phenomenon of the source-to-sink pathway that exists in all plants, in which carbohydrates are generated by source materials such as photosynthetic tissues, and mobilized to their final sink tissue, namely, the developing grain [11,49]. It therefore makes sense that the GN and GW are negatively linked as more grains represent more sinks to fill, frequently resulting in fewer assimilates making it into individual grains. This relationship is not one to one, and crosses between high GN and high GW lines often display transgressive segregation for the respective traits, as well as yield [8,9,10]. The continued reporting of transgressive segregation for such crosses indicates that GN and GW can still be improved independent of each other.

Improvements to the source side of the source-to-sink pathway appear to have a direct positive effect on GW, generally independent of the GN. By enhancing the source, the potential for more photo assimilates to be mobilized to developing grains becomes apparent. Enhancements to the source potential should have little effect on the GN, at least at the level of genetic control [53,54,55]; however, increased availability of the resources within the plant may ultimately lead to fewer floret abortions, allowing for an increase in the GN along with GW [53]. Several strategies have been highlighted for improving the source potential of the wheat plant, first and foremost being the plant size. Larger plants are associated with larger grains and plant size can be affected in a large magnitude by a small number of genes, such as *Rht* [55]; however, controlling the plant height is important in wheat breeding for different reasons other than control over the GW, such that it is not always advantageous to breed for the largest plants. The established positive relationship between the flag-leaf photosynthetic area and GW indicates that exploiting the genetic control over the flag-leaf size is a reasonable strategy for increasing the sink potential, and enhancing the GW without decreasing the GN [54]. Similarly, the association between higher stem WSC and a higher GW is apparent, such that the selection for lines with increased WSC represents an improvement to the plant source potential [53]. Utilizing awned genotypes as opposed to those without represents an increased investment into the source potential of the plant, as the awn photosynthetic tissue directly moves assimilates into the developing grains; however, the development of awns likely represents a competitive sink with developing maternal tissues and is associated with a decrease in the GN as a result [79,80,81,82]. While awns represent an increase to the source potential, they may simultaneously decrease the sink potential and are, therefore, a less useful trait in breeding for a stable yield increase. Selecting for awned versus awnless lines does have utility depending on the target environment.

Given that much of the gain in yield since the Green Revolution has come from increasing the GN per unit area [15], stable GW improvement is potentially a more fruitful direction to find the necessary yield improvements required to achieve the goal of doubled production by 2050 [1]. A major area of genetic control over GW comes in promoting more numerous and larger pericarp cells at the time of anthesis [59,60]. This can be thought of as improvements to the sink potential of the plant and there are many genetic pathways in the plant that can be manipulated to enhance the pericarp cell proliferation. Control over the G1/S phase genes can be exploited to promote pericarp cell proliferation [17,18,19,61]. Increasing both the cytokinin and expansin expression in the developing wheat head, leading up to the time of anthesis, also promotes larger and more numerous pericarp cells [37,40,41,42,62,63,69]. Allelic differences in the genes that control the synthesis and degradation of cytokinin and expansin in the wheat head leading up to anthesis can be exploited to promote pericarp cell proliferation. Increasing the cytokinin and expansin levels in the plant leading up to anthesis does more than just improve the sink potential of pericarp tissues, it also enhances the source potential of photosynthetic tissues given the association between increased cytokinin levels in the cell and increased chlorophyl production [34,63]. This may explain why experiments that increase cytokinin or expansin levels in the cell frequently see stable GW improvements without GN decreases. A proportional increase to the source potential, alongside the increase to the sink potential may be essential in stable yield improvements outside of the normal GW/GN trade-off (Figure 7).

Genes that control the floral architecture and floret fertility also contain the potential for a stable yield improvement. *WAPO1* directly impacts the floral architecture with a higher expression leading to more spikelets and a higher GN; however, this is coupled with a higher rate of floret abortion [44]. *GNI* directly impacts floret abortions regardless of the environment, with less functional alleles resulting in a larger GN [47,48]. Combining alleles of *WAPO1* that promote more spikelets with alleles of *GNI* that reduce floret abortion may be useful in further enhancing GN. It is still possible for a larger number of grains to have a smaller GW as a result of the source-to-sink relationship, but this decrease can potentially be mitigated by additionally selecting for alleles of genes that directly enhance GW through the promotion of larger pericarp cells at anthesis (Figure 8). The frequency with which crosses between a high GN and high GW cultivar adapted to an area display transgressive segregation [8,9,10], is a good indicator of a strategy to identify the genes that may lead to the most stable yield gains. The recommendations for breeding would be to perform such a cross with locally adapted cultivars and utilize a Genome Wide Association analysis on the resulting population to identify which genes are associated with the most improvement to GN or GW in the adapted environment.

Genotype by environment interaction plays a large role in determining the best method by which to pursue a yield enhancement [12,13,14]. There are a wide variety of growing environments around the world, but they can be aggregated into two distinctly different situations. There are irrigated environments, in which the nutrients and water necessary to grow and maintain the plant are always or more frequently available as opposed to dryland environments, where the resources are limited. Dryland or resource limited environments can, therefore, be thought of as limiting to the source potential of a plant, while irrigated environments are significantly less limited in this regard. Given this understanding, it would be more advantageous to focus on the development of source potential through genetic control when breeding for resource limited environments, and placing more focus on developing the sink potential through genetic control when breeding for irrigated environments. This idea is reinforced by the fact that high WSC lines are associated with more drought tolerance, as the increased abundance in WSC in the plant tissues can be mobilized for grain fill even during periods of reduced photosynthesis under water stress [57,58]. In resource limited environments it would be recommended to breed for lines with high WSC levels in the stem tissues that persist past anthesis into the grain fill. The source potential can be further increased by utilizing awned lines and selecting for a larger flag-leaf area [79,81]. Secondary to the efforts to improve the source potential of the plant at the genetic level, would be control of the size of the pericarp tissue through selection of the proper alleles of *TaGS5, TaGW2, TaDA1, TaExpA6*, and *TaCKX* genes (Figure 9); however, in a source limited environment, adjusting the potential of the sink may not result in a stable yield increase [16]. With that in mind, it would also seem appropriate to select for alleles of *WAPO1* that promote fewer spikelet formations, such as *WAPO-A1a*, and alleles of *GNI* that promote floret abortion, such as the wild type *GNI-105N* allele, so that in a source limiting environment, fewer sinks are competing for assimilates during the grain fill (Table 8).

Irrigated environments have significantly less source limitations, as the availability of water should drive photosynthesis consistently throughout the life cycle of the plant, provided there is sufficient fertilizer. While lines with increased WSC may result in a benefit to the yield in irrigated environments, the fact that these environments are significantly less source limiting decreases the potential impact of a high WSC line on the yield [53], suggesting that other breeding strategies may be more effective. With less source limitations, breeding for a higher GN still represents a potential for increasing the yield, therefore, strategies should include the selection of alleles of *WAPO1* that promote spikelet proliferation, such as *WAPO-A1b*, and alleles of *GNI* that decrease the floret abortion, such as *GNI-105Y*. Increases in GN should be coupled with the selection of alleles of *TaGS5, TaGW2, TaDA1, TaExp06*, and *TaCKX* genes that increase the GW [18,26,29,32,38,41] (Table 9). In particular, the alleles of *TaGS5*, *TaExp06*, and *TaCKX* genes that promote increased pericarp cell size and numbers would likely be most useful, as their mechanism for promoting a higher GW comes through the control of cellular division and elongation [37,41,61]. *TaGW2* and *TaDA1* also affect the cytokinin levels in cells, which in turn impact cellular division; however, they also impact the levels of AGPase expression [26,63]. Alleles of these two genes can be selected to promote a higher GW, but may increase the GW by mobilizing more starch to the grain [65]. This could affect the end-use quality of the grain, specifically the protein content, in such a way as to make the final grain less economically useful. More studies on how the allelic variation of *TaGW2* and *TaDA1* effects the protein content of the final grain are warranted in order to determine the best strategy for exploiting these loci for germplasm improvement.

The trade-off between GN and GW is often thought of as a simple competition for resources between sinks [5,6,7]. In many cases this is true, but this review has highlighted multiple strategies for making yield improvements through the control of one or both of these traits in spite of this trade-off. The most obvious strategy calls for proportionally improving the source potential of the plant alongside improvements to the sink potential of the grain. Crop scientists have been charged with making rapid yield improvements to meet the growing demand and with fixing the most beneficial alleles of *TaGS5*, *TaGW2*, *TaDA1*, *TaExp06*, *TaCKX6* and *GNI* into more modern cultivars that will go a long way in supporting this important endeavor.

## Figures and Tables

**Figure 1 plants-11-01772-f001:**
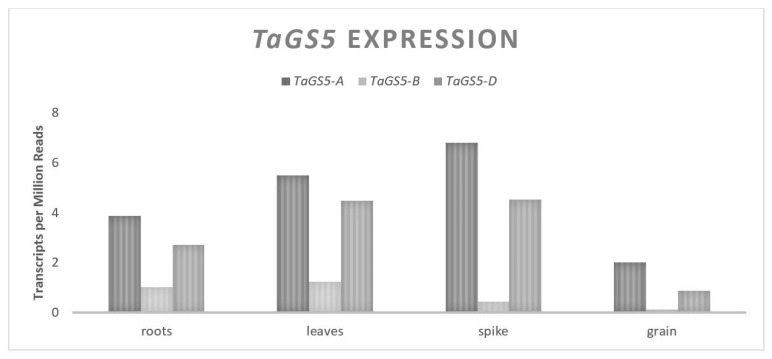
*TaGS5* relative homolog expression levels.

**Figure 2 plants-11-01772-f002:**
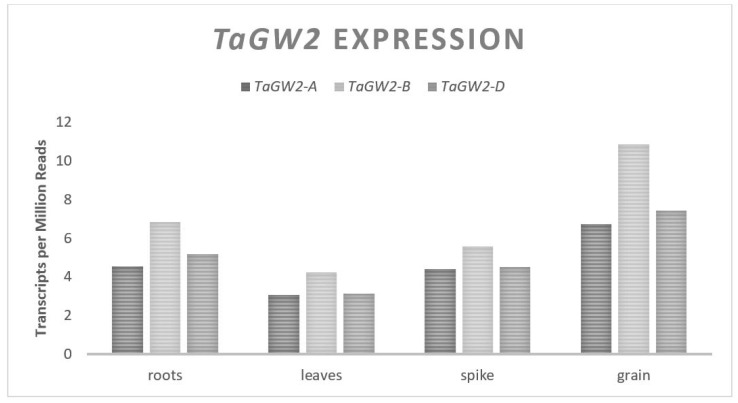
*TaGW2* relative homolog expression levels.

**Figure 3 plants-11-01772-f003:**
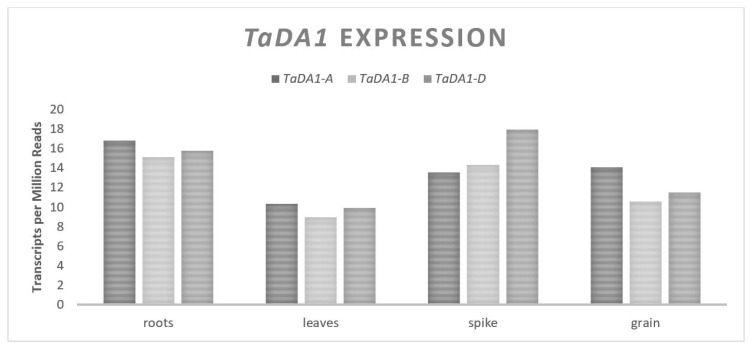
*TaDA1* relative homolog expression levels.

**Figure 4 plants-11-01772-f004:**
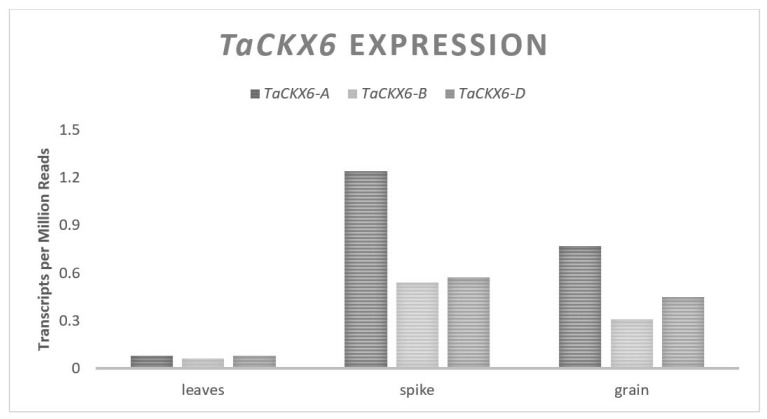
*TaCKX6* relative homolog expression levels.

**Figure 5 plants-11-01772-f005:**
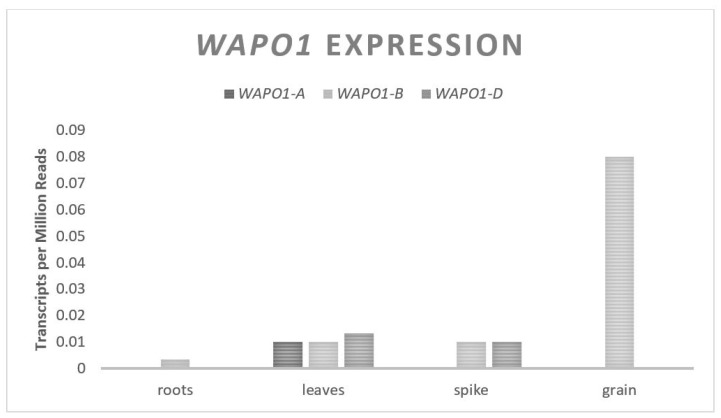
*WAPO1* relative homolog expression levels.

**Figure 6 plants-11-01772-f006:**
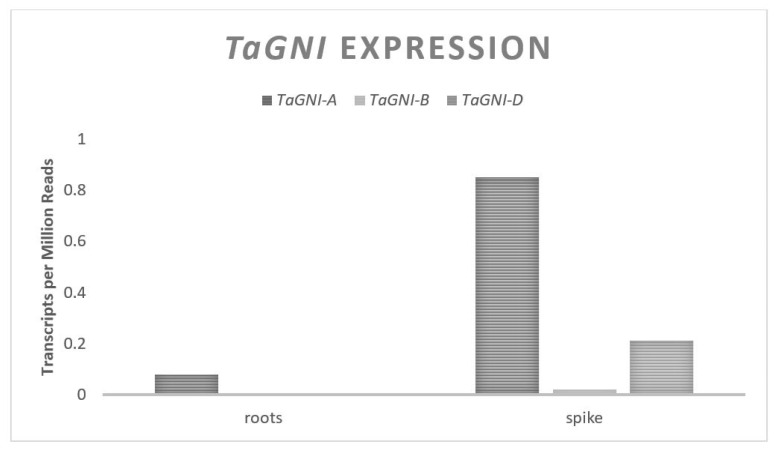
*TaGNI* relative homolog expression levels.

**Figure 7 plants-11-01772-f007:**
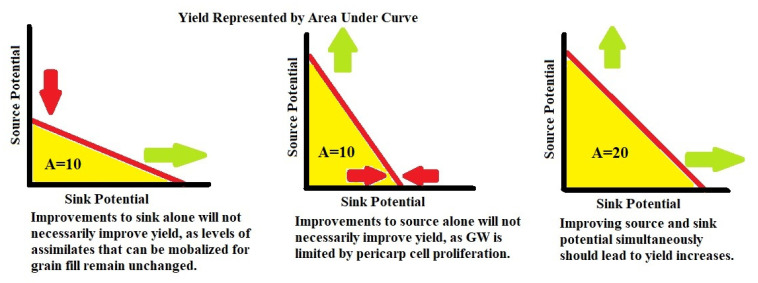
Rationale for Yield Improvement Strategy—yield is represented by A for area under the curve. Green arrows represent an increase in potential, while red represents a decrease or holding stable in potential.

**Figure 8 plants-11-01772-f008:**
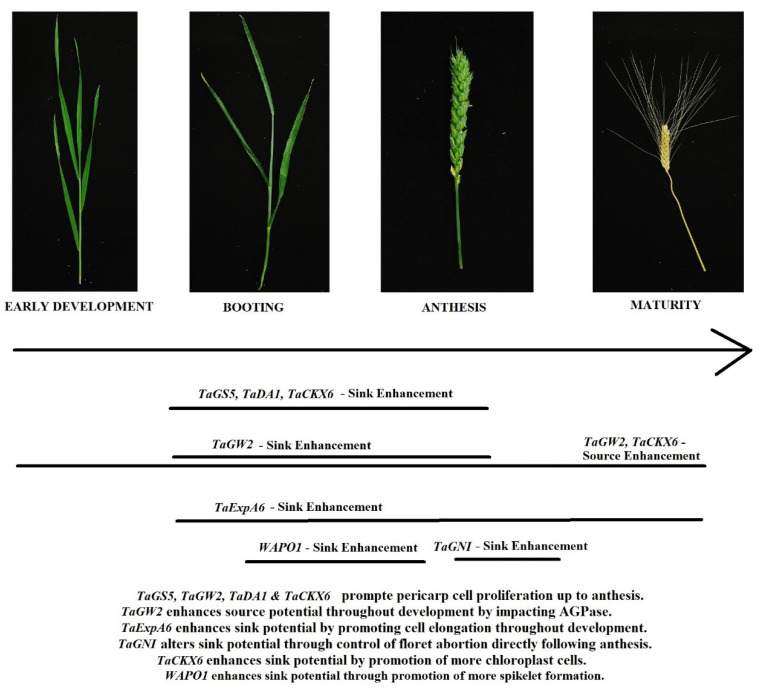
Timing and mechanism of impact for genes discussed. Images ordered by direction of wheat development. Black bars represent relative timing of gene impact to plant development.

**Figure 9 plants-11-01772-f009:**
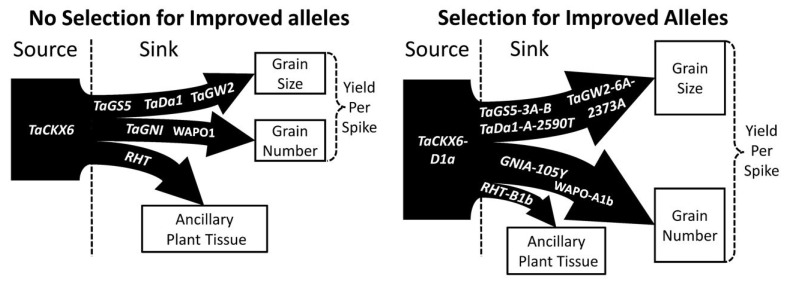
Selecting for increased source potential, grain size, grain number, and decreased ancillary plant tissue enables breeders to simultaneously improve yield without trade-offs between grain size and grain number. Genes impacting trait displayed in arrows. Thickness of arrows and source box represent improvement when selecting for beneficial alleles.

**Table 1 plants-11-01772-t001:** Genes discussed in review with major contributing references.

Gene	UniProt ID	Trait	Control	Chromosome Set	References
** *TaGS5* **	A0A2K9YUP7	GW	Positive	3S	[17,18,19]
** *TaGW2* **	A0A097ET43	GW	Negative	6S	[20,21,22,23,24,25,26,27,28,29,30]
** *TaDA1* **	A0A3B6BWR0	GW	Negative	2	[31,32,33]
** *TaCKX6-D1* **	I4AZT3	GW	Negative	3	[34,35,36,37,38,39]
** *TaExpA6* **	Q6QFA7	GW	Positive	4	[40,41,42]
** *WAPO1* **	A0A341Y3L0	GN	Positive	7L	[43,44,45,46]
** *TaGNI* **	A0A3B6U951	GN	Negative	2L	[47,48]

**Table 2 plants-11-01772-t002:** Natural Alleles of *TaGS5*.

Allele	Position	Domain	Type	Base	Gene Effect	GW	GN
** *TaGS5-3A-G* **	+2334	Exon 6	SNP/Missense	G	Reduced Function	↓	Unchanged
** *TaGS5-3A-T* **				T	Increased Function	↑	Unchanged
** *TaGS5-3A-A* **	−1925	Promoter	Single Base Insert	-	Reduced Expression	↓	Unchanged
** *TaGS5-3A-B* **				G	Increased Expression	↑	Unchanged

**Table 3 plants-11-01772-t003:** Natural alleles of *TaGW2*.

Allele	Position	Domain	Type	Base	Gene Effect	GW	GN
** *TaGW2-6A-494G* **	−494	Promoter	SNP	G	+Expression	↓	Not Reported
** *TaGW2-6A-494A* **				A	−Expression	↑	Not Reported
** *TaGW2-6A-2372G* **	+2373	Exon 5 Splice Acceptor Site	SNP	G	Normal	↓	Unchanged
** *TaGW2-6A-2373A* **				A	Truncated Protein	↑	Unchanged
** *TaGW2-6A-977* **	+977	Exon 8	Single Base Insert	−	Normal	↓	Not Reported
** *TaGW2-6A-977T* **				T	Truncated Protein	↑	Not Reported
** *TaGW2-6A1* **	−230/−117	Promoter	114 Base Deletion	+	+Expression	↓	↑
** *TaGW2-6A1del* **				−	−Expression	↑	↓

**Table 4 plants-11-01772-t004:** Natural alleles of *TaDA1-A*. GW and GN arrows represent trends in data but lack statistical significance in experimentation.

Allele	Position	Domain	Type	Base	Expression	GW	GN
** *TaDA1-A-2590T* **	+2590	Exon 3	SNP/Missense	T	−Expression	↑	↓
** *TaDA1-A-2590A* **				A	+Expression	↓	↑
** *TaDA1-A-4437G* **	+4437	Exon 7	SNP/Missense	G	Even Expression	↑	↓
** *TaDA1-A-4437A* **				A	Even Expression	↓	↑

**Table 5 plants-11-01772-t005:** Natural alleles of *TaCKX6-D*.

Allele	Position	Domain	Type	Base	Expression	GW	GN
** *TaCKX6-D1a* **	+3092/3109	Intron 2	18 Base Deletion	−	−Expression	↑	Not Reported
** *TaCKX6-D1b* **				+	+Expression	↓	Not Reported
** *TaCKX6a02-D1a* **	+3761/3790	3’ UTR	29 Base Deletion	−	Unknown	↑	Not Reported
** *TaCKX6a02-D1b* **				+	Unknown	↓	Not Reported

**Table 6 plants-11-01772-t006:** Natural alleles of *WAPO1*.

Allele	Position	Domain	Type	Base	Gene Effect	GW	GN
** *WAPO-A1a* **	+140	Exon 1	SNP/Missense	G	Reduced Function	↓	↑
** *WAPO-A1b* **				T	Increased Function	↑	↓
** *WAPO-A1a* **	−599/−485	Promoter	115 base indel	−	Reduced Expression	↓	↑
** *WAPO-A1b* **				+	Increased Expression	↑	↓

**Table 7 plants-11-01772-t007:** Natural alleles of *TaGNI-A*.

Allele	Position	Domain	Type	Base	Gene Effect	GN	GW
** *TaGNIA-105N* **	+349/351	Exon 2	SNP/Missense	AAC	Functional	↓	↑
** *TaGNIA-105K* **				AAG	Reduced Function	↑	↓
** *TaGNIA-105Y* **				TAC	Reduced Function	↑	↓

**Table 8 plants-11-01772-t008:** Breeding recommendations for dryland, source limiting environments. Focus is on improving source potential primarily, followed by improving sink tissue potential while reducing competition between sinks.

Dryland—Resource Limited—Source Limiting.				
Gene/Trait	Allele	GW	GN	Target
**Awned/Awnless**	Awned	↑	↓	Improve Source
**Water Soluable Carbohydrates**	High	↑	Even	Improve Source
** *TaGS5* **	*TaGS5-3A-B*	↑	Even	Improve Sink
** *TaGW2* **	*TaGW2-6A-2373A*	↑	Even	Improve Sink and Source
** *TaDA1* **	*TaDA1-A-2590T*	↑	↓	Improve Sink
** *TaCKX6* **	*TaCKX6-D1a*	↑	Not Reported	Improve Sink
** *WAPO1* **	*WAPO-A1a*	↑	↓	Reduce Sink Competition
** *TaGNI* **	*TaGNIA-105N*	↑	↓	Reduce Sink Competition

**Table 9 plants-11-01772-t009:** Breeding recommendations for irrigated environments. Focus on improving sink tissue potential primarily, followed by improvement to source potential with less concern for competition between sinks.

Irrigated—Resource Abundant				
Gene/Trait	Allele	GW	GN	Target
**Awned/Awnless**	Awnless	↓	↑	Improve Source
** *TaGS5* **	*TaGS5-3A-B*	↑	Even	Improve Sink
** *TaGW2* **	*TaGW2-6A-2373A*	↑	Even	Improve Sink and Source
** *TaDA1* **	*TaDA1-A-2590T*	↑	↓	Improve Sink
** *TaCKX6* **	*TaCKX6-D1a*	↑	Not Reporterd	Improve Sink
** *WAPO1* **	*WAPO-A1b*	↓	↑	Promote More Sink Tissue
** *TaGNI* **	*TaGNIA-105Y*	↓	↑	Promote More Sink Tissue

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
