# Peer review of "Genes Impacting Grain Weight and Number in Wheat (Triticum aestivum L. ssp. aestivum)"

_plants, 2022, doi:10.3390/plants11131772_

Round 1

Reviewer 1 Report

The current manuscript presents an interesting review about the genes related with the grain weight and number in common wheat. In general, the manuscript is well written and it can be useful for its potential readers in this journal.

Nevertheless, some minor changes should be carried out through the text.

The author use the term ‘wheat’ as generic, but they should change to ‘common wheat’ because only speaks of this species. Consequently, the title should be slightly change enclosed this expression together with the correct Latin name of this species: Triticum aestivum L. ssp. aestivum.

The other species named with the Latin names should be also corrected. The author indicates in L267: Z. mays, due this the first time, this should change to Zea mays L. Similar change should be carried out in L302, they write: ‘In Arabidopsis, DA1…’, but the correct should be: ‘In Arabidopsis thaliana (L.) Heynh., DA1…’

Author Response

We corrected the genus species names as suggested.

Reviewer 2 Report

The authors have reviewed the genes impacting the grain weight and number of grains in wheat.

I believe that the authors have provided sufficient and adequate background, reviewed the literatures in depth. The overall presentation of figures and tables are fine, though some modifications are needed, as I have suggested below. The conclusions are appropriate based on available data. I have now major concerns with this manuscript but a few minor editorial suggestions for the authors to consider if a revision is required by the editor:

Abstract: the authors have reviewed many aspects of grain weight and number but this information is not adequately addressed in abstract. I would expand the Abstract to include some specifics of the areas reviewed by the authors.

Table 1: remove the “underline” feature. Same problem with Tables 2 and 6.

Figures 1-5: resolution needs to be improved.

Figure 6: the arrows and color coding (red, green, or yellow) should be explained in the caption.

Figure 7: resolution needs to be improved; move the explanations of the functions of genes from the figure to the figure caption.

Table 7: remove the first roll and make the second roll as the first roll. Same problem for Table 8.

Figure 8: remove the dark background of the arrows.

Line 63, affect

Author Response

Fixed the underlining issue, increased resolution of figures. We tried removing the dark background of the arrows in the final figure but think it presents the information better as is.